# Ultrasound-Assisted Extraction of Applewood Polyphenols at Lab and Pilot Scales

**DOI:** 10.3390/foods12173142

**Published:** 2023-08-22

**Authors:** Hannes Withouck, Axel Paelinck, Imogen Foubert, Ilse Fraeye

**Affiliations:** 1Biochemical Innovation Team Odisee (BIT-O), Department Chemistry, University College Odisee, Gebroeders De Smetstraat 1, 9000 Ghent, Belgium; axel.paelinck@odisee.be; 2Meat Technology & Science of Protein-rich Foods (MTSP), Department M2S, Leuven Food Science and Nutrition Research Centre (LFoRCe), KU Leuven—Ghent, Gebroeders De Smetstraat 1, 9000 Ghent, Belgium; ilse.fraeye@kuleuven.be; 3Research Unit Food & Lipids, KU Leuven Kulak, E. Sabbelaan 53, 8500 Kortrijk, Belgium; imogen.foubert@kuleuven.be; 4Leuven Food Science and Nutrition Research Centre (LFoRCe), KU Leuven, Kasteelpark Arenberg 20, 3001 Leuven, Belgium

**Keywords:** UAE, mass–volume ratio, solvent composition, bioactive compounds

## Abstract

This study focused on the extraction of polyphenols from applewood using ultrasonic-assisted extraction (UAE). First, the influence of solvent composition and mass–volume (m:v) ratio on the extraction yield was studied at a lab scale (200 mL). Overall, a ratio of 1:33 (m:v) resulted in a higher yield of polyphenols. Furthermore, both a higher yield of polyphenols and higher antioxidant capacity were detected in the extracts produced in the presence of a 30 *v*/*v* % ethanol mixture compared to pure water; a further increase in ethanol did not improve the extraction yield. Second, under the optimal conditions (30 *v*/*v* % ethanol–water; 1:33 and 1:20 (m:v)), the UAE technique was applied at a pilot scale (100 L). At 1:33 (m:v), the polyphenol yield was lower at the pilot scale compared to the lab scale; by contrast, at 1:20 (m:v), production at the pilot scale resulted in a higher yield compared to the lab scale. To identify and quantify individual polyphenols, HPLC-PDA analyses were performed. Phloridzin appears to be the major identified compound. Finally, the UAE process was compared to a conventional solid–liquid extraction technique, showing that a significantly higher yield could be obtained with UAE.

## 1. Introduction

The European Green Deal strives for better protection of both people and the environment against harmful substances [1]. In addition, it envisions an efficient use of resources, with an emphasis on recovering waste streams while targeting sustainability. From this perspective, wood from apple trees that is perceived as useless may in turn become feedstock for the extraction of bioactive compounds by applying a green, environmentally friendly technique. Usually, after the harvesting season (December–January), apple trees that produce an insufficient amount of apples or apples of poor quality are used as firewood for heating or electric energy generation [2] and biochar [3]. The wood can also be buried in the soil, enabling the biodegradation of vegetal matter and reducing the need for fertilizers [4,5]. However, applewood contains significant amounts of bioactive compounds in the form of polyphenols, specifically in the classes of flavonoids, proanthocyanidins, and phenolic acids, which may supplant commonly used synthetic antioxidants in both foods and the pharmaceutical sector. To our knowledge, apart from our own studies [6,7,8,9], no other studies have focused on the extraction of bioactive compounds from applewood [10].

Despite the fact that conventional solid–liquid extraction (CE) techniques are widely used, these extraction protocols are generally very time- and energy-consuming. They require large volumes of expensive and pure solvents, and they have a low extraction selectivity and a high solvent evaporation rate during the process [11]. CE uses solvents (acetone, methanol, ethanol, etc.) in combination with heat and/or agitation to improve the mass transfer and solubility of bioactive compounds from plant materials [12]. The high solvent and energy consumption not only increases operating costs but also causes additional environmental problems [13], implying that these techniques are no longer identifiable with the ecological and environmentally friendly steps to be taken in today’s industry. Techniques responding to the current need for less energy and more efficient extraction include ultrasound-assisted extraction (UAE), microwave-assisted extraction (MAE), and pressurized liquid extraction (PLE), among others [14]. This study focuses on the use of UAE due to its improved extraction yield and short extraction time. Additionally, the technology is simple and relatively low-cost [14,15,16]. Rodriguez et al. [17] studied the effect of UAE on the recovery of polyphenols from olive pomace and observed a polyphenol yield 58% higher than that observed using CE. UAE extracts bioactive compounds using high-intensity sound waves [18,19]. Ultrasound has the ability to initiate cavitation, vibration, mixing, and other comprehensive effects in the solvent mixture contributing to the release and solubilization of bioactive compounds [16,20]. The cavitation, thermal, and mechanical effects caused by the UAE process lead to cell (wall) destruction, particle size reduction, and increased mass transfer, allowing the diffusion of a solvent into the crushed particles [21,22,23,24]. This effect is much stronger at low frequencies (18–40 kHz) and practically negligible at high frequencies (400–800 kHz) [20]. A number of reported applications have already demonstrated the potential of the UAE technology as a green, environmentally friendly technique for the recovery of bioactive compounds [23,25,26,27,28]. However, an ultrasonication system has its disadvantages, such as producing undesirable changes in molecules and requiring optimization [29,30].

Our previous study already proved the higher yield of the UAE technique compared to CE for extracting antioxidant polyphenols from apple-tree-derived bark and core wood [7]. In this 2019 study, an ultrasonic bath was used in a laboratory setting (200 mL), and both UAE and CE extraction were performed at 60 °C for 30 min. The impact of various solvents, including water, acetone, methanol, and ethanol at ratios of 20, 40, 60, 80, and 100% *v*/*v* (organic solvent/water) on the efficiency of the extraction process was examined. Extracts obtained using 40% to 80% of organic solvent exhibited the highest yield of polyphenols, with no significant difference observed between the respective solvents. These findings were consistent with those of other studies [31,32]. In general, alcohols with different levels of water are widely used to extract polyphenols from plant materials [19]. As ethanol is cheap, can be derived from a renewable source (sugar cane), and is classified as a GRAS (generally recognized as safe) solvent, it fits the green chemistry approach [15,33] and will be used in this study in different ratios with water.

The current study focuses on the potential of the UAE technique in an industrial context. To this end, specific UAE probes were used, which deliver a higher ultrasonic intensity, allowing for a reduction in the extraction time and temperature while enhancing the extraction yields [28,34,35,36,37].

Overall, the goal of the current study was to investigate the UAE of polyphenols from applewood both at the lab and at the pilot scale. More specifically, the study focuses on two specific objectives. First, extraction conditions were optimized at the lab scale. On the one hand, the effect of the ethanol–water ratio was studied. On the other hand, the ratio of the applewood mass to the solvent volume was varied in order to evaluate to what extent raw material throughput can be maximized while minimizing solvent use. The second goal included testing the UAE process, with optimal extraction conditions, at three different levels of extract production: 200 mL, 1000 mL, and 100 L. The process at the pilot scale (100 L) is an innovative approach as few studies have examined UAE on such a large scale [16]. The UAE process at a pilot scale was performed in order to evaluate the potential of this extraction technique for industrial applications, in response to the common criticism of UAE not being scalable [25]. Although the solvent and the temperature were consistent with the process at a lab scale, UAE-related parameters such as power, frequency, geometry, and mixing efficiency were different at the different scales, which may affect the obtained results [17].

For both goals, the generated extracts were evaluated in terms of total phenolic content (TPC) and total flavonoid content (TFC), as well as in terms of antioxidant activity using in vitro assays. Moreover, the characterization of the polyphenols was performed by high-pressure liquid chromatography equipped with a photodiode array detector (HPLC-PDA). These measurements evaluate the yield of total and individual polyphenols and antioxidant activity of the applewood extracts obtained. In addition, the yield of the UAE technique was compared with the CE technique, which is currently the most commonly used technique in industry. For industrial relevance, maximum extraction yield must be achieved. From this perspective, the residual polyphenol content in the pulp (residual fraction post-extraction) was also studied to examine the amount of polyphenols extracted using UAE and CE. This paper represents an important step in using UAE as an industrial process to recover bioactive compounds.

## 2. Materials and Methods

### 2.1. Chemicals

Aluminum chloride, ethanol 99%, sodium acetate, and sodium carbonate (anhydrous) were purchased from Chem-Lab (Belgium, Zedelgem). Acetic acid, 2,4,6′-tris(1-pyridyl)-5-triazine (TPTZ), hydrochloric acid, iron (II) sulfate heptahydrate, Folin–Ciocalteu reagent, quercetin (≥95%), gallic acid (GA) (≥97.5%), and DPPH were obtained from Sigma-Aldrich (Belgium, Overijse). Iron (III) chloride hexahydrate was purchased from Acros Organics (Belgium, Geel). Barium chloride, sodium dihydrogen phosphate, methanol, and the solvent for extraction, ethanol, were purchased from VWR Chemicals (Belgium, Leuven). Potassium acetate was purchased from Merck (Belgium, Hoeilaart). Sulfuric acid was obtained from Fisher Scientific (Belgium, Merelbeke).

The analytical reference compounds phloridzin (>99%), (-)-epicatechin gallate (≥98%), quercetin-3-D-galactoside (≥90%), (-)-epicatechin (≥97.9%), naringin (≥95%), ideain chloride (≥90%), vanillic acid (≥97%), gallic acid (≥97.5%), p-coumaric acid (≥98%), and caffeic acid (≥ 98%) for HPLC calibration were acquired from Sigma-Aldrich. Phloretin (≥99%), (+)-catechin (≥99%), kaempferol-3-O-glucoside (≥95%), avicularin (quercetin 3-α-L-arabinofuranoside,≥96%), kuromanin chloride (≥96%), chlorogenic acid (≥99%), t-cinnamic acid (≥95%), ferulic acid (≥90%), daidzein (≥99%), naringenin (≥99%), procyanidin B1 (≥90%), and B2 (≥90%) were acquired from Extrasynthese (France, Lyon). Ultrapure water was obtained from a Milli-Q System from Merck (Beglium, Hoeilaart).

### 2.2. Biological Material

Jonagold applewood samples, originating from three geographical locations in Belgium to create a representative mixture, were provided by Steps VOF (Gingelom, Belgium, 50°44′2″ N 5°12′35.2″ E), Waremme Fruit (Waremme, Belgium, 50°42′46.1″ N 5°15′44.4″ E), and Wolfcarius Fruit (Dentergem, Belgium, 50°94′70.5″ N 3°39′98.3″ E). The Jonagold variety currently dominates the apple production industry in Belgium and was therefore selected. Samples for the first experiment were collected in January 2020 after apple harvesting. For the second and third experiments, a second batch of applewood samples was gathered in January 2022. At each harvest site, a minimum of three apple trees were shredded. After 48 h of oven-drying at 37 °C, the wood from the three locations was ground to fine particles below 1 mm (FRITSCH Pulverisette 19), and a uniform mixture was generated which was immediately vacuum-sealed and stored at −80 °C until further analysis.

### 2.3. Experimental Setup

This section describes the overall experimental setup. Further experimental details on the extractions (UAE at lab and pilot scale, CE) and analyses (antioxidative potential and polyphenolic content) are provided in Section 2.4, Section 2.5, Section 2.6 and Section 2.7, respectively.

This study included three experiments.

The first experiment aimed to study the effect of the ratio of the applewood mass to the solvent volume (mass–volume ratio) and solvent composition on the extraction yield at a lab scale (200 mL) in order to create an optimized UAE extraction process. The mass–volume ratios were varied between 1:33, 1:20, and 1:10% (m:v), and the solvent compositions studied included pure water and water–ethanol mixtures (30, 50, and 70% (*v*/*v*)).

In the second experiment, the extraction process was conducted at three levels of extract production: 200 mL, 1000 mL, and 100 L. The second experiment expanded on the outcomes of the first experiment, based on which 30 *v*/*v* % ethanol as a solvent and two mass–volume ratios (1:33 and 1:20 (m:v)) were selected.

Extracts generated in experiments 1 and 2 were screened for both the levels of polyphenols via TPC, TFC, and the reducing power using the DPPH-RSA and FRAP assays, as described below. In the second experiment, characterization of the polyphenols was performed by HPLC-PDA.

In the third experiment, the yield of UAE was compared with CE. Both extraction techniques were conducted at the same scale (lab scale, 200 mL) and with the same solvent (30% ethanol) and m:v ratio (1:20). TPC analysis and HPLC screening were performed on each extract. In addition, the residual content of polyphenols in the pulp, the residual stream after extraction, was quantified in order to further evaluate the extraction yield. The remaining polyphenols in the pulp were quantitatively extracted by Soxhlet extraction and determined by the TPC method and by HPLC screening.

For all experiments, yields are expressed as a function of the content of dry wood (DW).

All extracts were independently produced in duplicate. On each extract, all analyses were performed at least in triplicate. This means that the obtained standard deviations not only incorporate the analytical error but also the extract preparation error.

### 2.4. Extraction Techniques

At the lab scale (200 and 1000 mL), ultrasound was applied to a mixture of applewood and solvent in an ultrasound lab device UP 200St (Hielscher Ultrasound Technology, Germany, Figure 1-left) via a continuous process. This device is specially developed for the sonification of volumes from 100 to 2000 mL. Additionally, a magnetic agitator (IKA C-MAG MS 7) inside the extraction vessel was essential for the homogenization of the mixture. The ultrasound sonotrode, a titanium alloy probe with a cylindrical diameter of 26 mm and length of 55 mm (S26d26), was placed inside a sound-reduction box. For both extraction scales, the probe was placed, according to the manufacturer’s guidelines, 27.5 mm below the surface of the wood–solvent mixture. The amplitude was set to 70%, generating a power of 50 W, and the sonification time was 10 min since preliminary experiments indicated that longer sonification times did not result in a higher yield. The frequency equaled 26.0 ± 1.0 kHz. The acoustic energy density (AED), calculated as the ratio between the ultrasonic power applied (P) and the extraction volume [28,37], was 225 ± 5 W/L for the 200 mL and 45.0 ± 0.1 W/L for the 1000 mL scale, respectively. The solvent temperature, which was monitored throughout the entire extraction process, was 25 ± 1 °C. To avoid oxidation, nitrogen was supplied during extraction with a flexible pipe along the bottom to maintain optimal percolation of the mixture. Extractions were performed at a pressure of 4 bar.

For the extraction at pilot scale (100 L), a UIP2000hdT (Hielscher Ultrasound Technology, Germany, Teltow) served as an ultrasound processor via a continuous process. Additionally, an agitator (IKA Eurostar 400) inside the extraction vessel was essential for the effective homogenization of the mixture (Figure 1, right). For this extraction, the probe was placed 200 mm below the surface of the wood–solvent mixture. The amplitude was set to 70%, generating a power of 950 W, the sonication time was 10 min, and the frequency equaled 20.0 ± 0.5 kHz. The acoustic energy density (AED) was 9.5 ± 0.1 W/L. The solvent temperature, which was monitored throughout the entire extraction process, was 25 ± 1 °C. Nitrogen was supplied with a flexible pipe along the bottom to maintain optimal percolation of the mixture. The nitrogen pressure was adjusted to the volume of the container (8 bar).

CE was conducted at lab scale (200 mL) with 30 *v*/*v* % ethanol and a ratio of 1:20 (m:v). Extraction was conducted for 10 min on a heating plate with mechanical agitation. The solvent temperature (60 ± 3 °C) was monitored throughout the entire extraction process. During extraction, nitrogen was supplied with a flexible pipe along the bottom to avoid oxidation and to maintain optimal percolation of the mixture. Extractions were produced at a pressure of 4 bar.

All extracts underwent centrifugation at 3005× *g* for 10 min (Eppendorf centrifuge 5810). After centrifugation and vacuum filtration, the supernatant was filtered through 0.45 µm polyvinylidene fluoride membranes (Phenomenex) and stored at −80 °C until further analysis.

In experiment 3, the pulp underwent a quantitative Soxhlet extraction according to the method described by Alara [38] with slight modifications, in order to determine to what extent polyphenols were still present in this residual stream and consequently remained unextracted.

### 2.5. Antioxidant Potential of Extracts

#### 2.5.1. DPPH-RSA Assay

The antioxidant activity of the extracts was measured by a modified test based on the DPPH-RSA (2,2-diphenyl-1-picrylhydrazyl radical scavenging activity) assay [8] and the FRAP (ferric reducing antioxidant power) assay [39].

DPPH is a stable free radical and is applied to test the antioxidant potential of polyphenols [8,32]. The DPPH-RSA assay consisted of adding 25 µL extract to 200 µL DPPH solution (prepared in 0.1 mmol/L ethanol). After 90 min incubation at room temperature in the dark, the absorbance was measured at 515 nm using ethanol as blank. The following formula was used to compute the percentage of inhibition or RSA:% inhibition= [(Ac − As)/Ac] × 100,
where Ac is the control reaction absorbance and As is the testing extract absorbance. The results are expressed as % inhibition as a function of the DW content.

#### 2.5.2. FRAP Assay

The FRAP assay, based on the reduction of a ferric complex (Fe^3+^-TPTZ) to the ferrous form (Fe^2+^-TPTZ) in the presence of antioxidants, was performed by mixing 10 µL of extract with 300 µL of FRAP reagent (300 mmol/L acetate buffer—pH 3.6, 10 mmol/L TPTZ (2,4,6-tris (1-pyridyl)-5-triazine) in 40 mmol/L HCl and 20 mmol/L FeCl_3.6_H_2_O in a ratio of 10:1:1). After an incubation of 15 min at 37 °C, the absorbance was measured at 593 nm. Aqueous solutions of FeSO_4.7_H_2_O in a range from 200 to 1000 µmol/L were used for generating a calibration curve. FRAP values were expressed as mmol of ferrous equivalent per gram DW (mmol Fe(II)/g DW).

The measurements of both the DPPH-RSA and FRAP assays were conducted on a microplate reader (Reader EPOCH 2T monochromator).

### 2.6. Phenolic Content

#### 2.6.1. Total Phenolic Content

A modified total phenolic content (TPC) assay [8], based on the original Folin–Ciocalteu (FC) method [40], was performed. The reaction mixture consisted of 25 µL of sample or standard solution, 75 µL of distilled water, and 25 µL FC reagent (previously diluted 10 times with water). After 6 min, 100 µL of sodium carbonate 7.5% (m:v) was added. Absorbances at 740 nm were measured after an incubation period of 120 min in the dark. The results are expressed as gallic acid (GA) equivalents per g DW (mg GAE/g DW) using a calibration curve established by a GA standard solution (10–100 µg/mL). The measurements were performed on a microplate reader (BioTech EPOCH 2).

#### 2.6.2. Total Flavonoid Content

The principle of total flavonoid content (TFC) is based on the formation of a stable complex by aluminum chloride with the C4 keto group and the C3 or C5 hydroxyl group of the flavones and flavonols [41] and performed according to Moreira [8] and Paz [42]. The assay consisted of adding 100 µL of deionized water followed by 10 µL of NaNO_2_ (50 g/L) and 25 µL of standard, sample, or deionized water (blank) into a microplate. After 5 min in the dark, 15 µL of AlCl_3_ (100 g/L) was added and, after 1 min of reaction in the dark, 50 µL of NaOH (1 mol/L) was also added. Finally, the microplate was introduced into the microplate reader (BioTech EPOCH 2), and after incubating in the dark for 10 min and 10 s of shaking at medium velocity, absorbance was recorded at 510 nm. An epicatechin standard solution (300 µg/mL) was prepared to set up the calibration curve (7.5–225 µg/mL). Consequently, the TFC value is expressed as mg epicatechin equivalents per g DW (mg EE/g DW).

### 2.7. HPLC-PDA Analysis

In order to achieve a reliable overview of the type and quantity of polyphenols, the individual polyphenols present in the extracts obtained in experiments 2 and 3 were determined by HPLC-PDA as described in an earlier study [7], with modifications. In short, this analysis was performed on a Shimadzu HPLC system consisting of a low-pressure quaternary gradient unit (model LC-20AT) with an in-line degasser (model DGU-20A5R) and an auto-sampler (model SIL-20AT). The system was equipped with a column oven (model CTU-20AC) and a PDA detector (model SPD-M20A High-Performance Liquid Chromatography PDA detector). A Phenomenex Gemini C18 column (250 mm × 4.6 mm, 5 μm) and a guard column with the same characteristics maintained at 40 °C were used for the separation of polyphenols. Mobile phase A (HPLC-grade methanol) and mobile phase B (ultra-pure water), both with 0.1% formic acid, were used for elution at a flow rate of 1.0 mL/min, with a gradient as described in [7]. The injection volume for the samples and reference compounds was 20 µL. All extracts were filtered through a 0.45 µm polyvinylidene fluoride membrane (Phenomenex) and mixed with the internal standard (daidzein, 50 ppm) before analysis. Calibration curves, with an internal standard (daidzein, 50 ppm) from the standards (mentioned in the chapter chemicals) prepared in a methanol–water mixture (50:50, *v*/*v*) by dilution of appropriate amounts of the stock solutions, were obtained. The identification of the polyphenols in the obtained extracts was based on the comparison of the retention times and the spectral data of each compound with those of pure standards injected under the same conditions. The content of individual identified polyphenols in each extract was expressed as mg of compound per g DW (mg phenolic compound/g DW). Relevant analytical data, namely the regression equation, R^2^, the limit of detection (LOD), and quantification (LOQ) are shown in Appendix A.

### 2.8. Statistical Analysis

Each extract was independently produced in duplicate. All analyses were performed on each extract at least in triplicate. As such, the obtained standard deviations not only incorporate the analytical error but also the extract preparation error. Values are expressed as mean ± standard deviation (SD). Data analysis was carried out with the IBM SPSS Statistics 23 software. A significance level of *p* < 0.05 was employed in all tests.

For experiment 1, the obtained data were statistically processed with a two-way between-groups analysis of variance (ANOVA), with the m:v ratio (1:33, 1:20, and 1:10 (m:v)) and solvent composition (0, 30, 50, and 70% (*v*/*v*)) as the independent variables. Similarly, for experiment 2, a two-way ANOVA was conducted, with the extraction volume (200 mL, 1000 mL, and 100 L) and the m:v ratio (1:33 and 1:20) as the independent variables. For both experiments 1 and 2, a multiplicative model was applied to verify whether interaction effects occurred. In the case of significant interaction, the interaction term was further interpreted using one-way ANOVA followed by a post hoc Tukey’s honestly significant difference (HSD) test. For experiment 3, comparing the extraction yield between UAE and CE, statistical analysis was performed by an independent samples *t*-test.

## 3. Results

### 3.1. Influence of Solvent Composition and Mass–Volume Ratio on Extraction Yield

Extraction optimization is required in order to identify the optimal attributes for the efficient extraction of natural compounds. As stated in the introductory section, this study focused on ethanol–water mixtures of varying compositions. In addition, the effect of the mass–volume ratio was studied, as from an economic and industrial point of view processing higher masses of waste stream with minimal use of solvents would be beneficial.

The impact of the solvent mixture and m:v ratio on the yield (mg/g DW) of polyphenols and the antioxidant power is studied and presented in Figure 2. There was a significant interaction between the effects of the m:v ratio and solvent mixture for all attributes measured. Therefore, one-way ANOVA was applied to the interaction term to compare the results statistically. It was observed that the m:v ratio significantly affects the phenolic content (Figure 2A,B) and antioxidant power (Figure 2C,D). The levels of polyphenols and reducing power generally tend to decrease as the m:v ratio increases. The 1:33 (m:v) ratio showed significantly higher results for most tested attributes compared to 1:20 and 1:10 (m:v). This observation was also made in a study on the UAE extraction of total flavonoids from Osmanthus fragrans Lour. flowers [43]. In this study, the extract yield also decreased as the m:v ratio increased from 1:40 to 1:10 (m:v). This study equally showed that a ratio lower than 1:40 (m:v) did not result in a higher yield of flavonoids. However, the effect of the m:v ratio on extraction yield in the literature was inconsistent, as a study on the total phenolics from grapes showed no significant differences between the results from experiments using the 1:20 and 1:10 (m:v) ratios [44]. The extraction yield was, however, lower using a 1:5 (m:v) ratio. Another study [45] on the extraction of phillyrin from Forsythia suspensa showed that the 1:33 and 1:20 (m:v) ratios resulted in a decrease in extraction yield compared to 1:10 (m:v). It is understood from these observations that the m:v ratio can affect the extraction yield of the UAE process in different ways. In general, a larger solvent volume can dissolve constituents more effectively, leading to an improvement in the extraction yield [46]. A higher concentration difference of the solute increases diffusion and consequently dissolution of the solute in the solvent, improving the extraction process. Furthermore, if the m:v ratio is too high, the viscosity of the solution may be high, which may impose limitations on the cavitation effect [36]. The increase in the contact area between the material and solvent may also increase the yield. By contrast, a very low m:v ratio can also lead to an enhanced cavitation effect resulting in the degradation of the desirable solute itself [19]. Therefore, it is important to choose an optimal m:v ratio for the ultrasonic extraction process, which depends on the type of sample, the type of compounds extracted, and the extraction solvent being used.

The spectrophotometric tests in this study indicate that 1:33 (m:v) results in a higher yield compared to 1:20 and 1:10 (m:v). From an economic perspective, using a large amount of solvent is not considered cost-effective due to the high operating cost of solvents and energy consumption. Therefore, in addition to 1:33 (m:v), a m:v ratio of 1:20 (m:v) was also selected for experiment 2, in which extract production was performed both at lab and pilot scale.

Looking at the influence of the solvent in more detail, it was observed that the addition of water to ethanol, creating a more polar medium, facilitated the extraction of polyphenols. The extraction of polyphenols, measured by TPC, was favored by using 30% (*v*/*v*) ethanol compared to pure water. When using the specific test for flavonoid determination (TFC), a ratio of 1:33 (m:v) in combination with 30% (*v*/*v*) ethanol showed significantly higher amounts compared to almost all other conditions (Figure 2B). When pure water was used for the extraction of polyphenols from applewood, it can be observed that approximately 35% less polyphenols, measured by TPC and TFC, could be extracted compared to extracts produced with ethanol–water mixtures (30% (*v*/*v*)). Using higher amounts of ethanol did not result in further improvement. This trend has also been noted in the extraction of flavonoids from grapefruit solid waste [18] and hawthorn seed [47]. Those studies also stated that an increase in ethanol concentration increased the yield of polyphenols until a certain ethanol concentration, while further increasing the concentration even had a negative effect.

In general, the results of the first experiment suggest the use of 30% (*v*/*v*) ethanol as the most appropriate solvent. Apart from a high yield, this composition also offers economic and environmental advantages, as only 30% ethanol is required.

### 3.2. Influence of UAE Production Scale

Based on the observations of the first experiment, extractions in experiment 2 were carried out with m:v ratios of 1:33 and 1:20 in the presence of 30% (*v*/*v*) ethanol. This experiment intended to monitor the effect of scale (200 mL, 1000 mL, and 100 L) and m:v ratio (1:33 and 1:20) on the yield determined by TPC, TFC, and the polyphenolic composition by HPLC-PDA. The antioxidant activity of the obtained extracts was also measured. There was a significant interaction between the effects of the m:v ratio and extraction scale for all attributes measured. Therefore, one-way ANOVA was applied to the interaction term to compare the results statistically.

Comparing the yield of UAE performed at lab scale (200 mL and 1000 mL) with the extraction performed at pilot scale (100 L), higher yields of TPC and TFC were obtained at lab scale when a m:v-ratio of 1:33 was applied (Figure 3A,B). The reducing power of applewood extracts measured by FRAP (Figure 3C) and DPPH-RSA assays (Figure 3D) showed the same tendency. These observations were confirmed by the obtained HPLC data (Figure 4). In the presence of 1:33 (m:v), extract production at the pilot scale had a slightly lower yield compared to the lab scale. This probably relates to the lower acoustic energy density (AED) applied at the pilot scale compared to the lab scale. A different generator is employed in the production of extract at lab and pilot scales due to the volume to be treated. For both generators, an amplitude of 70% was set, which generated power of 50 and 950 W for the lab and pilot setups, respectively. The AED for extractions at the pilot scale in this study was 9.5 W/L, which is substantially lower compared to AED used at 200 and 1000 mL (225 and 45 W/L, respectively). The higher AED may hence explain the higher yield of polyphenols for extractions produced at a 200 mL scale. Still, it is worth noting that despite decreasing power density at the pilot scale, the extraction yield remained good.

A review of the literature concerning the UAE of plant materials revealed that authors often only focus on the amplitude or the nominal power output of the UAE system and do not report the total treated volume, which means that the AED cannot be calculated, in spite of its importance in the performance of the UAE process. This complicates the comparison between experimental results [25,34,48].

Extracts produced with a 1:20 (m:v) ratio demonstrated an opposite trend in comparison with 1:33 (m:v), as extracts produced at pilot scale tend to show slightly higher yields in TPC, TFC, and higher reducing power measured by the FRAP assay (Figure 3). A similar trend is observed when studying the results of HPLC analysis (Figure 4): at a 1:20 (m:v) ratio, extracting at the pilot scale showed a positive effect on the extraction yield of polyphenols compared to the lab scale. Using a 1:20 (m:v) ratio at the pilot scale resulted in yields at least 30% higher in comparison to extracts produced at the lab scale. This implies that the extraction process at a larger scale, combined with a higher m:v ratio, shows beneficial effects in extracting polyphenols from applewood. In fact, the results are rather surprising considering a lower power density is deployed at the pilot scale. However, it must be noted that limited homogenization was observed if a ratio of 1:20 (m:v) was used during extraction at the lab scale. The ground wood always tended to settle out unlike in the case of 1:33 (m:v). At the 100 L scale, a good homogeneity occurred at both 1:33 and 1:20 (m:v) ratios. This may explain why at a ratio of 1:20 (m:v) a slightly higher extraction yield was obtained at the pilot scale compared to the lab scale.

Overall, it can be concluded that high yields of polyphenols were obtained at the pilot scale. Therefore, UAE is considered to have promising potential for application in the industrial extraction of polyphenols from applewood.

### 3.3. Polyphenolic Profile

Figure 5 exhibits a representative PDA chromatogram. As can be seen in the chromatogram, almost all major peaks were identified. Seven peaks remained unidentified, although they represented only a minor proportion of the total phenolic compounds. Table 1 presents the amounts of identified polyphenols relative to the total polyphenol content detected by HPLC-PDA for all extracts produced as part of experiment 2. Dihydrochalcone phloridzin, a glycoside of phloretin already found in apple leaves [49], appears to be the major compound identified in all produced extracts. Its level accounts for almost 60% of the total amount of determined polyphenols. The limited number of studies performed on apple tree residues already established that the apple tree (Malus sp.) accumulates high amounts of phloridzin [7,50,51]. In the present study, the highest content of phloridzin was 13.23 ± 0.44 mg/g DW, which is in the same range as the value reported by Moreira [8]. However, as opposed to the present study, the latter study examined different parts of the apple tree, i.e., the root, bark, and core wood with 23.40 ± 1.17, 21.66 ± 1.08, and 7.45 ± 0.18 mg/g DW respectively. The reported value is also similar to the one reported in apple tree leaf extracts (21.07 ± 0.06 mg/g DW) [52]. The study by Xü [53] showed that the aglycon form of phloridzin, phloretin, is present in the bark of apple trees. In the present study, small amounts of phloretin were observed in extracts produced at a pilot scale with 1:20 (m:v) (0.19 ± 0.01 mg/g DW), which were close to the values reported by Rana [52] (0.13 ± 0.07 mg/g DW), and Moreira [8] (0.08 ± 0.01, 0.12 ± 0.01, and 0.28 ± 0.01 mg/g DW for core wood, bark, and root, respectively). Both phloretin and phloridzin have the potential to trap reactive dicarbonyl species and therefore inhibit the formation of advanced glycation end products which exert direct toxicity on cells and tissues [54]. Next, they also showed an inhibitory effect against the oxidation of omega-3 polyunsaturated fatty acids and fish oil [55,56]. Besides the chalcones, the four main flavonoids identified in the majority of the extracts include kaempherol-3-O-glucoside, naringin, (-)-epicatechin, and avicularin. In the case of kaempherol-3-O-glucoside, naringin, and (-)-epicatechin, the contribution to the total amount of polyphenols corresponds to 18.6–21.1%, 4.2–4.6%, and 3.0–3.8%, respectively. Compared to the studies by Rana and Moreira [8,52], the amounts of (-)-epicatechin extracted from applewood are higher. Avicularin, a quercetin O-glycoside in which an alpha-L-arabinofuranosyl residue is attached at position 3, represents between 7.4 and 8.4% of the identified polyphenols.

Phenolic acids are hardly found in applewood extracts. The identified and quantified phenolic acids are chlorogenic and p-coumaric acids. Comparing these results with the available literature, the same tendency is noted in apple tree leaves [52,57]. However, the lower number of identified phenolic acids is rather surprising since a larger number of phenolic acids is found in oak, pine, spruce, fir, walnut, etc. [58]. Yet the hydroxycinnamic acids detected in applewood appear to be available in greater quantities than in spruce and fir. The most common benzoic acids found in the bark of woody plants are vanillic, gallic, syringic, and protocatechuic acids. The most common cinnamic acids are p-coumaric, caffeic, ferulic, and synaptic acid [59].

Finally, procyanidins, polymerized forms of flavanols, were detected in the applewood extract and represent approximately 2.4–2.7% of all detected polyphenols. Procyanidins have been reported to exhibit broad benefits to human health and are used in the prevention of cancers, cardiovascular diseases, diabetes, etc. [60].

The present study aimed to determine whether extract production at different scales affects the ratio of polyphenols present. Chlorogenic acid was significantly affected by both the applied m:v ratio and scale. Extracts produced with a 1:20 (m:v) ratio contained a slightly higher proportion compared to the extracts produced with 1:33 (m:v). Furthermore, extracts produced at the pilot scale show higher amounts compared to extracts produced at the lab scale. By contrast, phloridzin, procyanidin B1, and B2 were not significantly affected by the scale or the m:v ratio. Coumaric acid showed an interaction effect, although no significant differences were observed from the one-way ANOVA results. For phloretin, (-)-epicatechin, and kaempherol-3-O-glucoside, in the extract production at pilot scale, using 1:33 (m:v), some small but statistically significant differences were observed. For avicularin and epicatechin gallate, the opposite was observed. In summary, although some statistically significant effects of scale, m:v ratio, and/or their interaction were observed, it can be concluded that the relative proportions of the different polyphenols were overall hardly affected by the scale and the m:v ratio of the extraction process. To our knowledge, no other studies have been conducted on the influence of extract production scale on the polyphenolic profile. This result was expected, given that the polyphenolic profile is highly dependent on the used solvent, which was identical for all extracts in experiment 2, namely 30% (*v*/*v*) ethanol.

### 3.4. Efficiency of the Ultrasound-Assisted Extraction Technique Compared to Conventional Solid–Liquid Extraction

The effectiveness of the optimized UAE technique compared to a CE technique was studied in more detail. The CE was conducted on a heating plate with a resulting solvent temperature of 60 ± 3 °C, while in the UAE extraction, it was 25 ± 1 °C. Based on the results of experiment 1, 30% (v/v) ethanol was selected as the solvent. For the UAE, a specific probe was used. The current study shows that the yield of polyphenols, recovered from applewood and measured through both TPC and HPLC, was higher via the UAE technique compared to the CE technique (Table 2). It can therefore be concluded that UAE has a positive influence on polyphenol extraction compared to CE.

In order to gain a more accurate understanding of the polyphenol extraction efficiency, the content of polyphenols in the residual stream (pulp) was determined. When the UAE technique was applied, there were hardly any polyphenols present in the residual stream, in contrast to the polyphenols found in the pulp after CE. These findings suggest that UAE is a more efficient process for extracting polyphenols from applewood compared to CE. This could be explained by the cavitation force of ultrasound, which can accelerate the mass transfer rate and facilitate the release of extractable compounds in the form of polyphenols [61].

## 4. Conclusions

This study focused on the application of UAE for the extraction of polyphenols from applewood. At the lab scale, the highest yields in polyphenols were obtained in the presence of ethanol–water 30% (*v*/*v*) and a m:v ratio of 1:33. The yields were significantly higher compared to the CE methodology.

When comparing the extractions performed at lab scale (200 mL and 1000 mL) to those at pilot scale (100 L), for a 1:33 (m:v) ratio, extract production at the pilot scale resulted in a slightly lower yield, while the opposite was observed at a 1:20 (m:v) ratio. Overall, a high polyphenol extraction yield was obtained on pilot scale. The reported results demonstrate that the proposed extraction technique has potential on a semi-industrial scale without compromising the polyphenol yield. UAE is a more efficient extraction technique compared to CE based on the polyphenol extraction yield, and the potential for production at pilot scale, all in addition to the advantages mentioned in the literature section. These findings confirm the potential of UAE as a future ecological extraction technique that meets the urgent demand for green extraction.

The polyphenols extracted from applewood may offer a sustainable source of bioactive compounds or antioxidants. HPLC analysis showed that the applewood extracts consisted mainly of phloridzin, kaempherol-3-0-glucoside, and avicularin, a quercetin O-glycoside; the compounds corresponded to 60, 20, and 8% of the identified polyphenols, respectively. These polyphenols could be used in the food and pharmaceutical sector.

## Figures and Tables

**Figure 1 foods-12-03142-f001:**
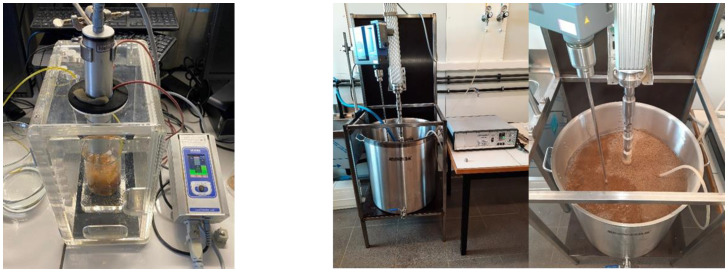
Ultrasound setup at lab scale (**left**) and at pilot scale (**right**).

**Figure 2 foods-12-03142-f002:**
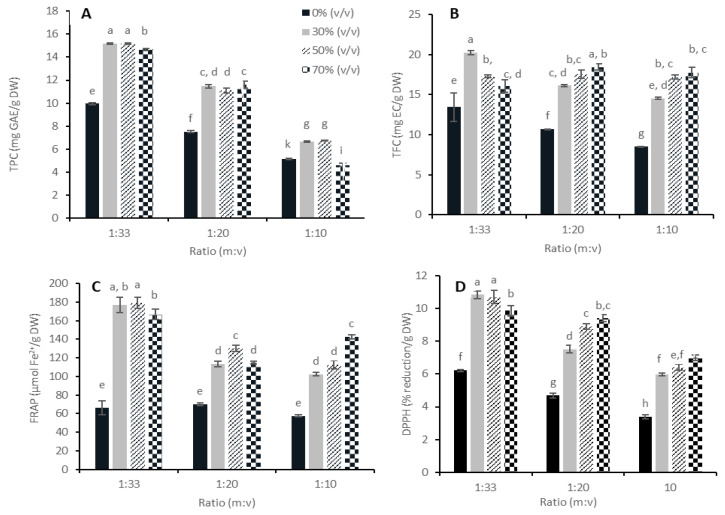
(**A**) Total phenolic content (TPC, results expressed in mg gallic acid equivalents/g dry wood), (**B**) total flavonoid content (TFC, results expressed in mg epicatechin equivalents/g dry wood), (**C**) ferric reducing antioxidant power (FRAP, results expressed in µmol Fe^2+^/g dry wood), and (**D**) 2,2-diphenyl-1-picrylhydrazyl radical scavenging activity (DPPH-RSA, results expressed in mg Trolox equivalents/g dry wood) of applewood extracts obtained by ultrasonic-assisted extraction (UAE) at 200 mL scale with ratios of 1:33, 1:20, and 1:10 (m:v) and in the presence of 0% (*v*/*v*—pure water), 30% (*v*/*v*—ethanol/water), 50% (*v*/*v*—ethanol/water), and 70% (*v*/*v*—ethanol/water). Error bars represent 1 standard deviation. The data were compared using one-way ANOVA followed by a post hoc Tukey’s test. Data with different lowercase letters are significantly different (*p* < 0.05).

**Figure 3 foods-12-03142-f003:**
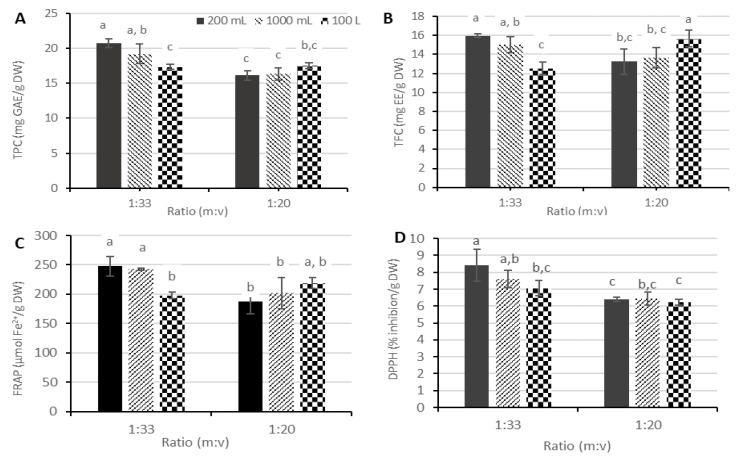
(**A**) Total phenolic content (TPC, results expressed in mg gallic acid equivalents/g dry wood), (**B**) total flavonoid content (TFC, results expressed in mg epicatechin equivalents/g dry wood), (**C**) ferric reducing antioxidant power (FRAP, results expressed in µmol Fe^2+^/g dry wood), and (**D**) 2,2-diphenyl-1-picrylhydrazyl radical scavenging activity (DPPH-RSA, results expressed in mg Trolox equivalents/g dry wood) of applewood extracts obtained by ultrasonic-assisted extraction (UAE) at the 200 mL, 1000 mL, and 100 L scales with ratios of 1:33 and 1:20 (m:v) in the presence of 30% (*v*/*v*—ethanol/water). Error bars represent 1 standard deviation. The data were compared using one-way ANOVA followed by a post hoc Tukey’s test. Data with different lowercase letters are significantly different (*p* < 0.05).

**Figure 4 foods-12-03142-f004:**
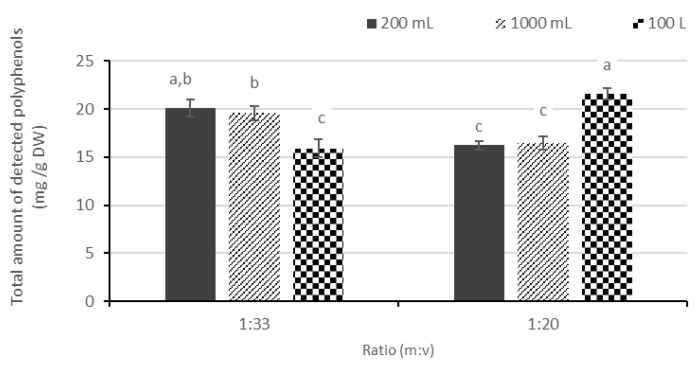
Total amount of identified polyphenol compounds with HPLC-PDA of applewood extracts obtained by ultrasonic-assisted extraction (UAE) at the 200 mL, 1000 mL, and 100 L scales with ratios of 1:33 and 1:20 (m:v) and in the presence of 30% (*v*/*v*)—ethanol/water. Results are expressed as mean ± standard deviation (n = 6). Error bars represent 1 standard deviation. The data were compared using one-way ANOVA followed by a post hoc Tukey’s test. Data with different lowercase letters are significantly different (*p* < 0.05).

**Figure 5 foods-12-03142-f005:**
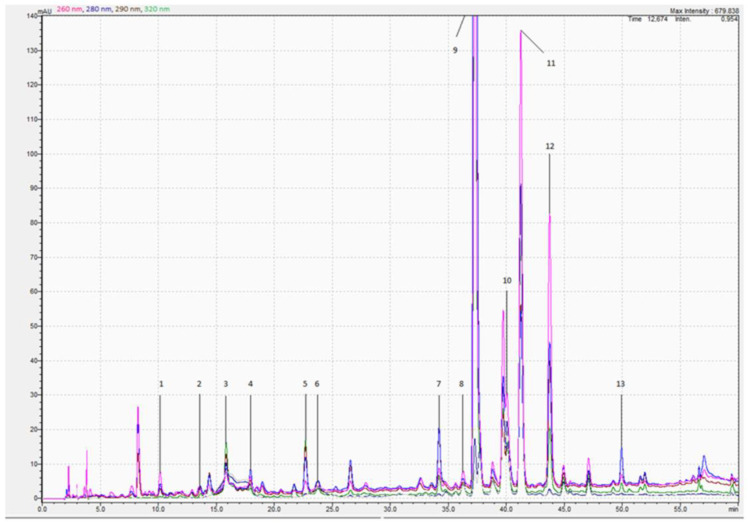
HPLC chromatogram of identified polyphenol compounds with HPLC-PDA at 260 (pink), 280 (blue), 290 (brown), and 320 (green) nm of applewood extracts obtained by ultrasonic-assisted extraction (UAE) produced at a 100 L scale with a ratio of 1:20 (m:v) and in the presence of 30% (*v*/*v*); (1) procyanidin B1, (2) procyanidin B2, (3) chlorogenic acid, (4) (-)—epicatechin, (5) p-coumaric acid, (6) epicatechin gallate, (7) naringin, (8) Quercetin 3-D-galactoside, (9) phloridzin, (10) avicularin, (11) kaempherol 3-O- glucoside, (12) daidzein, and (13) phloretin. A total of 7 peaks remained unidentified.

**Table 1 foods-12-03142-t001:** Relative content of the identified polyphenols in applewood extracts obtained by UAE performed at lab scale (200 and 1000 mL) and pilot scale (100 L) in the presence of 1:33 and 1:20 (m:v) ratios. Results are expressed as mean ± standard deviation (the amounts of identified polyphenols relative to the total polyphenol content, n = 6). A multiplicative model was applied to the individual polyphenols to verify whether interaction effects occurred. A significance level of *p* < 0.05 was considered statistically significant. In the case of significant interaction, the interaction term was further interpreted using one-way ANOVA followed by a post hoc Tukey’s test. Data with different lowercase letters are significantly different (*p* < 0.05). NS, nonsignificant.

Phenolic Compound	1:33 (m:v)									1:20 (m:v)									2-Way ANOVA Significance (Alpha = 0.05)		
(Relative %)	200 mL			1000 mL			100 L			200 mL			1000 mL			100 L			Scale	Ratio	Scale × Ratio
Kaempherol 3-O- glucoside	21.05% ^a^	±	0.61%	18.62% ^b^	±	0.35%	19.50% ^b^	±	0.50%	18.58% ^b^	±	0.25%	18.77% ^b^	±	0.09%	18.89% ^b^	±	0.95%	-	-	0.000
Quercetin 3-D-galactoside	0.063% ^a^	±	0.005%	0.055% ^a,b^	±	0.006%	0.050% ^b^	±	0.001%	0.050% ^b^	±	0.001%	0.055% ^a,b^	±	0.006%	0.060% ^a^	±	0.001%	-	-	0.000
Avicularin	7.44% ^b^	±	0.15%	7.71% ^b^	±	0.45%	7.61% ^b^	±	0.23%	7.42% ^b^	±	0.23%	8.36%^a^	±	0.07%	7.71% ^b^	±	0.05%	-	-	0.026
Flavonons	28.55% ^a^	±	0.77%	26.39% ^b,c^	±	0.81%	27.16% ^b^	±	0.73%	26.05% ^c^	±	0.48%	27.18% ^b^	±	0.17%	26.66% ^b,c^	±	1.00%	-	-	0.000
(-)-Epicatechin	3.64% ^a^	±	0.15%	2.99% ^b^	±	0.15%	3.03% ^b^	±	0.17%	3.33% ^a,b^	±	0.54%	3.75% ^a^	±	0.06%	3.65% ^a^	±	0.06%	-	-	0.001
Epicatechin gallate	0.74% ^b^	±	0.05%	0.73% ^b^	±	0.02%	0.74% ^b^	±	0.01%	0.85% ^a^	±	0.03%	0.72% ^b^	±	0.02%	0.74% ^b^	±	0.02%	-	-	0.000
Flavan-3-ols	4.38%^a^	±	0.20%	3.72% ^b^	±	0.17%	3.77% ^b^	±	0.18%	4.18% ^a,b^	±	0.57%	4.47% ^a^	±	0.08%	4.35% ^a^	±	0.08%	-	-	0.003
Naringin	4.15% ^b,c^	±	0.12%	4.39% ^a,b,c^	±	0.11%	4.46% ^a,b^	±	0.17%	4.62% ^a^	±	0.26%	4.08% ^c^	±	0.09%	4,39% ^a,b,c^	±	0.04%	-	-	0.000
Flavanons	4.15% ^b,c^	±	0.12%	4.39% ^a,b,c^	±	0.11%	4.46% ^a,b^	±	0.17%	4.62% ^a^	±	0.26%	4.08% ^c^	±	0.09%	4.39% ^a,b,c^	±	0.04%	-	-	0.000
Phloretin	0.61% ^d^	±	0.06%	0.67% ^c^	±	0.01%	0.78% ^b^	±	0.03%	0.84% ^a,b^	±	0.03%	0.85% ^a^	±	0.03%	0.87% ^a^	±	0.02%	-	-	0.000
Phloridzin	57.25%	±	1.16%	59.74%	±	0.50%	58.76%	±	0.95%	58.68%	±	0.31%	58.14%	±	1.11%	57.94%	±	1.70%	NS	NS	NS
Dihydrochalcones	57.86%	±	1.22%	60.41%	±	0.51%	59.54%	±	0.98%	59.52%	±	0.34%	58.99%	±	1.14%	58.81%	±	1.72%	NS	NS	NS
P-coumaric acid	0.48% ^a^	±	0.02%	0.49% ^a^	±	0.01%	0.47% ^a^	±	0.02%	0.47% ^a^	±	0.02%	0.48% ^a^	±	0.01%	0.49% ^a^	±	0.01%	-	-	0.040
Chlorogenic acid	2.41%	±	0.29%	2.14%	±	0.17%	2.37%	±	0.29%	2.45%	±	0.20%	2.32%	±	0.10%	3.06%	±	0.07%	0.001	0.000	NS
Hydroxycinnamic acids	2.89%	±	0.31%	2.63%	±	0.18%	2.84%	±	0.31%	2.92%	±	0.22%	2.80%	±	0.11%	3.55%	±	0.08%	0.002	0.000	NS
Procyanidin B1	0.45%	±	0.04%	0.43%	±	0.04%	0.50%	±	0.03%	0.40%	±	0.04%	0.45%	±	0.09%	0.44%	±	0.09%	NS	NS	NS
Procyanidin B2	1.90%	±	0.06%	2.26%	±	0.15%	2.23%	±	0.23%	2.18%	±	0.05%	2.12%	±	0.04%	2.09%	±	0.60%	NS	NS	NS
Proanthocyanidins	2.35%	±	0.10%	2.69%	±	0.19%	2.73%	±	0.26%	2.58%	±	0.09%	2.57%	±	0.13%	2.53%	±	0.69%	NS	NS	NS

**Table 2 foods-12-03142-t002:** Content of identified polyphenols in applewood extracts obtained by UAE and CE performed at lab scale (200 mL scale) in the presence of 30% (v/v) ethanol–water mixture and a 1:20 (m:v) ratio. Results are expressed as mean ± standard deviation (mg of compound/g DW, n = 6). The data were compared using an independent samples *t*-test.

Extraction Technique	TPC (mg GA/g DW)		HPLC (mg Marker Compounds/g DW)	
	Extract	Pulp	Extract	Pulp
UAE	16.95 * ± 0.31	0.05 * ± 0.06	16.22 * ± 0.64	0.00 *
CE	14.40 * ± 0.58	3.48 * ± 0.02	10.07 * ± 0.46	4.22 ± 0.01 *

* Significant difference (*p* < 0.05) in the amount of polyphenols available in pulp or extract treated by CE versus UAE extraction.

## Data Availability

All data are contained within the article.

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
