# Peer review of "Ultrasound-Assisted Extraction of Applewood Polyphenols at Lab and Pilot Scales"

_foods, 2023, doi:10.3390/foods12173142_

Round 1
Reviewer 1 Report
Dear Editor
The manuscript titled " Ultrasound-assisted extraction of apple wood polyphenols at lab 2 and pilot scale”. was completely revised and the following observations were found.
Introduction:
Line 42 – 44: The authors should mention the concentrations of polyphenols reported to date, as well as the specific families of polyphenols quantified.
Line 44: The authors should focus only on the food and pharmaceutical sector
Line 47 – 49: The authors should mention which are the conventional extraction techniques that have been developed, as well as what are the times, composition of solvents and volumes that were used.
Line 55 – 56: The authors should justify why ultrasound is better compared to other technologies, showing performance data in the extraction of polyphenols.
Line 58 – 65: Although UAE possesses advantages in extracting polyphenols, it is important to acknowledge the potential adverse effects associated with the utilization of high frequencies, temperatures exceeding 50°C, and extended process durations.
Line 84 – 91: Why do the authors use the ultrasound probe length as the only parameter?
Line 92 – 102: The objective of the study is not clearly defined in the introduction
Methodology
rearranging the methodology in a strict order to ensure clarity and avoid confusion for the reader. The order of the steps should be adjusted to follow a logical sequence: chemicals, raw material, extraction, analysis, and statistics. On the other hand, the mass-volume ratio must be expressed in ratios. For example, 1:10 (m:v)
Line 126 – 133: The work will be conducted in two distinct stages: the first stage involves a team operating at the laboratory level, while the second stage involves a team operating at scale. However, the authors should explain each stage separately, providing details about the equipment used, and including relevant photographs and diagrams.
Line 135 – 145: The TPC, TFC, DPPH, FRAP and polyphenols profile analyses should be mention in detail and mentioning the equipment used for these measurements. In addition, what were the analytical characteristics of the chromatographic analysis like LOD, LOQ, %RSD, regression equation, R2?
Line: 146: How was the sample conditioned prior to extraction?
Line 191 – 196: Why 30% ethanol in the EC? What is the reference?
Line 201 – 204: What were the conditions in Soxhlet extraction?
Line 210 – 217: For DPPH, what were the controls (blank and negative) used?
Line 128: los autores mencionan que optimizaran la UAE, pero no mencionan ningún tipo de metodología de optimización.
Line 276 – 291: The authors should separate the statistical analyses for each extraction stage. On the other hand, Why the tukey and t-student tests?
Results and discussions
Line 300 – 305: If the authors evaluate two study factors, why do they use a one-way ANOVA?
To ensure better comprehension, it is recommended to present the results using proportions. For example, Treatment A allowed to recover 20% more polyphenol content compared to Treatment B.
To enhance the results and discussion, it is important to compare the results with findings from other authors who utilized similar matrices. Additionally, it is necessary to provide an explanation for both the similarities and differences observed. Furthermore, it is crucial to discuss how the study factors influenced the responses.
It is not appropriate to write conclusions in the results and discussions section
Authors could consider comparing laboratory-level UAE versus pilot-scale UAE
The authors should explain why the UAE is better than the CE
Conclusions
The conclusion should be improved according to the previous recommendations.

Author Response
Dear reviewer,
First of all, we want to thank you for the very useful comments on our manuscript “Ultrasound-assisted extraction of apple wood polyphenols at lab and pilot scale”. These comments were helpful to further improve the manuscript. Hereby, we submit the revised paper.
We made several amendments in order to reply to your concerns. The amendments are explained in the appendix of this letter.
Sincerely,
The authors

Reviewer 2 Report
The main question addressed by the research is to investigate the ultrasound for extraction of apple wood polyphenols at lab and pilot scale. The polyphenols were extracted by conventional extraction and UAE at pilot and lab scale. The research design is appropriate and the article is well-written. However, the novelty, compared to other articles reported by the same authors is questionable.
The pilot scale UAE is novel, but the lab scale was already done at similar extraction parameters in previous research by same authors. It should be considered to update the research with more ultrasound extraction parameters, such as different amplitudes, different power and extraction time. Why did you choose 75% amplitude, 50 W and 10 min if it wasn't analysed in previous studies as the best extraction parameters? It would be better if you analysed different parameters and detected the optimal one for the comparison with pilot-scale.
Beside the articles published by authors before, there are no other articles and the topic is novel and interesting.
The figure 2 should be uploadad again with better quality.
The paper is easy to read, the concept is clear and the conclusions are consistent with the results and evidence presented.
Acknowledge the limitations of your study.
Author Response

(The authors gave the same response as above.)

Reviewer 3 Report
Article entitled Ultrasound-assisted extraction of apple wood polyphenols at lab and pilot scale. However, several things can be added for the improvement of the manuscript.
1. In the introductory section, the urgency of research should be explained more clearly, especially regarding extraction on a lab and pilot scale. Apart from that, the weaknesses and strengths of each extraction scale need to be explained in the introduction.
2. Style Figures 1, 2, and 3 should be consistent and increase the resolution.
3. The discussion regarding data phenomena in each table should be more in-depth and comprehensive.
4. The results and discussion section should be supplemented with a sub-chapter that discusses the correlation between observed parameters.
5. Conclusions from research results should be shorter and answer the research objectives.
6. The cited references should be more updated, at least for the last 5-8 years.
Moderate editing of English language required
Author Response
Dear reviewer,
First of all, we want to thank you for the very useful comments on our manuscript “Ultrasound-assisted extraction of apple wood polyphenols at lab and pilot scale”. These comments were helpful to further improve the manuscript. Hereby, we submit the revised paper.
We made several amendments in order to reply to your concerns. The amendments are explained in the appendix of this letter.
We hope these alterations are in accordance with the expectations.
Sincerely,
The authors

Reviewer 4 Report
The reviewed manuscript focuses on the ultrasound-assisted extraction of apple wood polyphenols at the laboratory and pilot scale. The research is interesting, but some of the results are questionable due to the unclear description of the analytical techniques employed. The authors used the HPLC PDA method for qualitative and quantitative profiling of polyphenols. They claim to have provided comprehensive information on the qualitative and quantitative composition of the polyphenolic compounds. They state that they utilized UV spectra and retention times for compounds identification. In my opinion, these data are not sufficient. In one chromatographic system, many compounds can have similar retention times, and structurally similar compounds may exhibit similar, non-diagnostic UV spectra. Additionally, no information was provided regarding the proportion of chromatographic peaks that were identified. Please include an example chromatogram and indicate how many compounds were present on the chromatogram and how many were identified. Furthermore, it is unclear what technique was used for quantitative determinations, whether an external standard method was employed or as the authors claim, an internal standard method (line 268). If an internal standard method was used, please provide details of the methodology. After clarifying these issues, the acceptance of the manuscript can be considered.
Author Response

(The authors gave the same response as above.)
